# Local DNA shape is a general principle of transcription factor binding specificity in *Arabidopsis thaliana*

Janik Sielemann [1,2,3], Donat Wulf [1,2,3], Romy Schmidt [4] & Andrea Bräutigam [1,2,3]✉

Understanding gene expression will require understanding where regulatory factors bind genomic DNA. The frequently used sequence-based motifs of protein-DNA binding are not predictive, since a genome contains many more binding sites than are actually bound and transcription factors of the same family share similar DNA-binding motifs. Traditionally, these motifs only depict sequence but neglect DNA shape. Since shape may contribute non-linearly and combinational to binding, machine learning approaches ought to be able to better predict transcription factor binding. Here we show that a random forest machine learning approach, which incorporates the 3D-shape of DNA, enhances binding prediction for all 216 tested *Arabidopsis thaliana* transcription factors and improves the resolution of differential binding by transcription factor family members which share the same binding motif. We observed that DNA shape features were individually weighted for each transcription factor, even if they shared the same binding sequence.

[1] Computational Biology, Center for Biotechnology (CeBiTec), Bielefeld University, 33615 Bielefeld, Germany. [2] Computational Biology, Faculty of Biology, Bielefeld University, 33615 Bielefeld, Germany. [3] Graduate School DILS, Bielefeld Institute for Bioinformatics Infrastructure (BIBI), Bielefeld University, 33615 Bielefeld, Germany. [4] Plant Biotechnology, Bielefeld University, 33615 Bielefeld, Germany. ✉email: andrea.braeutigam@uni-bielefeld.de

Changes in gene expression during development and invoked by environmental perturbations are critical to organismal function and these changes are influenced by DNA-binding transcription factors (TFs). *Arabidopsis thaliana* encodes 1533 DNA-binding TFs[1] many of which occur in protein families of a few to over a hundred members[2]. Gene expression of a particular gene is a complex read-out based on the presence of TFs and their spacing on the DNA, chromatin status, histone marks and presence of co-activators or repressors. Improving the understanding of those regulatory relations and pathways is necessary to tackle current agricultural challenges[3]. Hundreds of sequence motifs to which TFs bind have been characterised[4,5], but currently it is impossible to look at a promoter and understand its regulatory syntax. DNA is a very constrained molecule since its phosphate sugar backbone runs antiparallel while its bases are paired and arranged in rungs on a helical ladder. However, despite the constraints, the exact position of each base pair and each base in a pair is influenced by its surrounding bases. The pairs can be tilted, shifted, slid, rolled, risen and twisted relative to each other (Fig. 1d;[6,7]). The bases in a pair can be buckled, sheared, stretched, twisted, opened and staggered (Fig. 1d;[6]). The width of the minor groove is also influenced by the surrounding bases[8]. This DNA shape has been demonstrated to influence protein–DNA binding, for instance, of the *Drosophila* Scr Hox Protein[8,9] and the *S. cerevisiae* bHLH proteins Cbf1 and Tye7[10].

Many members of a particular TF family bind the same motif[4]. The *A. thaliana* genome contains 74 members of the WRKY TF family[11] regulating diverse processes from trichome and seed development to roles in biotic and abiotic stresses[12]. All WRKYs analysed bind to a consensus motif, the W-box, which is characterised by the TTGAC pentamer followed by C or T[13]. TFs of the 133 member bHLH family bind DNA via basic amino acids at the N-terminal end of the bHLH domain and bind a variation of the motif CANNTG, frequently the so-called G-box CACGTG[14]. This G-box motif is also bound by many bZIP TFs whose core motif is ACGT, the central nucleotides of the G-box CACGTG[15,16]. ChIP-seq data clearly indicates that only a subset of potential binding sites are indeed occupied at any given time in a particular tissue[17–19]. We hypothesised that DNA shape was a critical element for determining TF specificity within a TF family.

In amplified DNA affinity purification sequencing (ampDAP-seq) experiments amplified DNA devoid of methylation marks is bound to an in vitro produced TF and sequenced. For motif detection in ampDAP-seq or ChIP-seq, the DNA sequences bound by the TF are mined by motif search algorithms such as MEME or MEME-ChIP, which identify overrepresented motifs among the sequences[20]. For many TFs, biochemical experiments such as electrophoretic mobility shift assays (EMSAs) have confirmed that the motif is necessary for binding[21–23]. However, the comparison between the measured binding events from DAP-seq data and the frequency of the derived motif in the genome indicates that during motif prediction information is lost (Fig. 1a). In our analysis, the identified binding motif occurrence is on average 14-fold higher than the number of verified binding events (Supplementary Fig. 1). We hypothesised that binding specificity of a particular TF is encoded in DNA shape. To decipher the predictive power of DNA shape regarding protein binding, we trained a random forest model. This approach enables the detection of non-linear relationships between the shape of the bases and DNA-binding affinity. In addition, it allows the capture of possibly important combinatorial information of non-adjacent bases. We hypothesised that machine learned models trained on DNA shape within and surrounding the binding motif recover the lost information during motif

detection and generally improve prediction for TF binding in *A. thaliana*. In this work, we contribute to the understanding of protein–DNA recognition and demonstrate that DNA shape features enable a robust prediction of binding affinity regarding randomly generated motif containing sequences. In addition, we show that the models, trained on DNA shape, improve the distinguishability of binding locations for TFs that share the same binding motif. Understanding TF binding as a combination of motif sequence and motif shape brings us closer to predicting gene expression directly from sequence.

## Results

**DNA shape features explain large part of protein–DNA binding affinity.** To generate the datasets necessary for training, test, and validation, for each TF the sequence-based binding motif (henceforth called "core motif") was determined with MEME-ChIP using all ampDAP-seq peaks. Sometimes a motif is reported based on only the 600 peaks with the largest height[4]; we opted to capture all binding events. The genome was scanned with the motif generating two classes of events: motifs, which are not underneath a peak and hence not bound and motifs underneath a peak which were bound. Peak height is taken as a proxy for affinity. If a motif based on only the top 600 peaks was used, the number of sequence-only-based potential binding sites was increased (Supplementary Fig. 2) likely because those larger motifs reach the threshold for FIMO[24]-based extraction more easily compared to smaller motifs. A random forest decision tree (RF)-based regressor[25] was trained for each TF on the raw binding data using the peak height in ampDAP-seq as a proxy for binding affinity after binding data was filtered for single motif occurrences. On average, 146,326 sequences, which contain the binding motif, were extracted to train the models. For the TF with the least amount of training data we extracted 18,210 sequences, whereas the largest dataset for a TF contained 640,292 sequences. To ensure consistent 3D structure learning, the sequences were reverse complemented if the binding sequence was located on the reverse strand. We split the motif occurrences into a training and a validation set[25] using the measured signal value within the peak calling of the ampDAP-seq experiments as the numeric label. The training dataset was again split into train and test set (ratio 4:1) while performing cross-validation. In addition, we explored different ratios of train to test set (4:1, 3:1, 2:1 and 1:1) and observed no difference in performance (Supplementary Fig. 3). This indicates that the size of the input dataset is sufficient for robust training.

In total, 216 individual models for 216 TFs were generated. In each case, the shape-based predictor outperformed the motif search, based on the area under the precision recall curve (AUPRC). AUPRC improved between 2.8% and 362.7%, with an average of 93.2% (Fig. 1b). 33 TFs reach AUPRC of more than 0.8 indicating that the motif plus shape information suffices for prediction (Fig. 1b). 101 TFs show medium AUPRC between 0.5 and 0.8. The remaining 82 TFs show improved AUPRC compared to motif alone but does not exceed 0.5 (Fig. 1b). To investigate the influence of dimensionally reduced input features on model performance, all models were additionally trained after performing PCA on the shape features (Supplementary Fig. 4). The prediction performance was substantially lower when the models were trained on the dimensionally reduced features rather than the direct shape features. This observation also implies that the features are not considerably redundant.

Prediction of binding improved for all TF families, however, some families increased in prediction precision more than others (Supplementary Figs. 5 and 6). We analysed whether this discrepancy could be explained by the different dataset sizes,

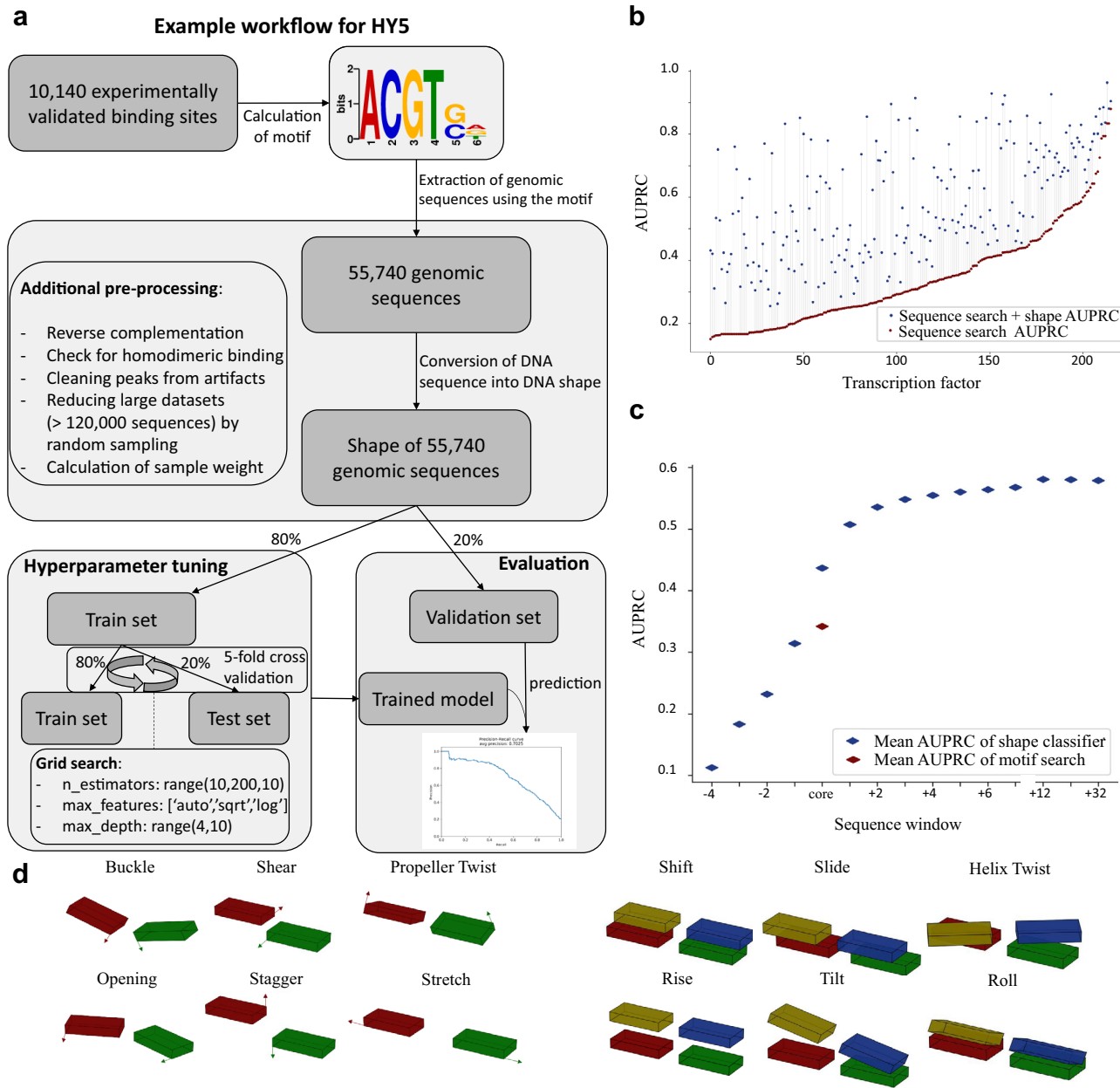

**Fig. 1 Overview of workflow and performance of shape-based binding site identification. a** Example workflow illustrating the computational steps from publicly available data to trained models capable of predicting protein–DNA binding affinity. **b** Performance of the random forest classifier using the border width (number of upstream and downstream bases) with the highest area under the precision recall curve (AUPRC) for each TF. **c** AUPRC for differing sequence widths. The width was increased upstream and downstream of the core motif sequence, respectively. **d** The different DNA shape features which were considered to analyse TF specificity. A query table was used for shape calculation[6].

which is the number of genomic sequences containing the motif (Supplementary Fig. 7). Indeed, we observed a slightly negative correlation between performance and dataset size with an $R^2$ of 0.154. Hence, on average the prediction performance is slightly worse for TFs whose sequence motif is more abundant in the genome. An additional comparison between ampDAP-seq and ChIP-seq data was performed for five TFs, for which data of both experimental procedures were available (Supplementary Fig. 8). We observed that ampDAP-seq outperformed ChIP-seq for each TF with an average of 98.6% higher AUPRC. This observation is in line with our expectation, as ampDAP-seq identifies binding events independent from conditions in the cell and uses unmethylated naked DNA for the identification of binding sites, whereas ChIP-seq captures the binding events in specific in vivo conditions. To test the contribution of shapes surrounding the motif, the amount of sequence, and therefore shape information, given to the regressor was varied and the training was repeated. The major contribution of shape information was localised to the core motif plus two bases on each side of the motif (Fig. 1c). These adjacent bases influence the shape of the bases and base pairs in the core[6]. Beyond the core motif shape, the information gain quickly levelled (Fig. 1c and Supplementary Fig. 9).

Two additional machine learning approaches were evaluated and compared with RF performance (Supplementary Fig. 10). The baseline neural network implementation performed overall slightly worse than the gradient boosting and random forest implementation. This relation would likely change with dedicated hyperparameter tuning. However, to be able to test the

importance of the different DNA shape features, we chose the RF-based machine learning approach to enable reliable feature extraction[26].

The models generated by the shape-based regressor show which shapes are important to the binding for each of 216 TFs tested as shown in the source data file. To test if any shape, shape type (intra-base pair vs. inter-base pair) or any position contribute a larger amount of information to the binding, the top five features were extracted for each TF. The 3D configuration of bases within the core binding sequence occupied 68% of the top five feature positions (Supplementary Fig. 11) as expected (Fig. 1c). Intra-base pair shapes contributed 39% and inter-base pair shapes contributed 61% (Supplementary Fig. 11) of the top five shapes within the core motif. Outside of the motif, the proportions reversed since intra-base pair shapes contributed 72% and inter-base pair shapes contributed 28% (Supplementary Fig. 11). Further outside of the core motif, the shear feature was overrepresented among the top five features (Supplementary Fig. 11).

**Improved resolution of differential binding by transcription factor family members.** If DNA shape predicts binding better than the motif alone and shape information used by TFs is varied, the prediction algorithm should be able to distinguish the binding between two TFs, which are predicted to bind the same motif sequence. To test this hypothesis, the models for TF pairs with the same sequence-only-based binding motifs were analysed.

The ERF/AP2 TFs CBF4 (AT5G51990) and ERF036 (AT3G16280) both bind the GTCGGT/C motif which occurs 31,155 times in the *A. thaliana* genome. According to ampDAP-seq, both TFs have 9910 binding sequences in common (Fig. 2a). ERF036 binds 2581 sequences which are not bound by CBF4, and CBF4 binds 6996 sequences not bound by ERF036. Using the published motif derived from the top 600 binding events results in a smaller overlap but leads to 103,779 extracted genomic sequences (Supplementary Fig. 12). To test whether the shape indeed encodes specificity, binding vs. non-binding was predicted by the models (Fig. 2b).

The shape information of the 31,155 genomic sequences, allows the regressor models to distinguish the binding events between the two TFs (Fig. 2b, c), even though the core sequence is the same and the majority of sequences are bound by both TFs according to ampDAP-seq. Each Venn diagram in Fig. 2c shows the distribution of binding sites applying the cut-off represented by the dashed line. For ERF036, 2162 out of the 2581 uniquely bound sequences were correctly identified as binding sequences, whereas only 416 out of the 6996 sequences bound uniquely by CBF4 were wrongly predicted as binding sequences (Fig. 2c). In total, the number of false-positive binding sequences from the motif search dropped from 18,664 (11,668 + 6996) to 1384 (968 + 416), which is an improvement of 93% less false-positive predictions when using the RF model. Likewise, for CBF4 84% of uniquely bound sequences were correctly predicted and only 15% of sequences bound uniquely by ERF036 are predicted as false positives (Fig. 2c). Here, the total improvement regarding false positives amounts to 86%, as the number of false-positive predictions dropped from 14,249 to 2038. To identify the features which contribute specificity to each TF, we extracted feature importances using 'shapley additive explanations' (SHAP)[26] (Fig. 2d). The outputs of the regressor models are influenced by different features. For ERF036, the slide at position -1 relative to the motif and the helix twist at position 5 in the motif is most influential, whereas for CBF4 the minor groove width at position 6 and the helix twist at position −1 contribute most to the decision of the RF model. This observation underlines that the

TFs, even though binding to the same core motif, are dependent on different peculiarities regarding the shape of the DNA (Fig. 2). These results are not family specific since TFs of the NAC family binding to the C(G/T)TNNNNNNNAAG motif (Fig. 2e, f), TFs of the WRKY family binding to TTGAC(T/C) motif, TFs of the bZIP family binding to ACGTCA motif and TFs of the C2H2 family binding to TTGCTNT motif show similar results (Supplementary Figs. 13–15). In summary, the features defined by the shape-based regressor are able to explain differential binding of two TFs binding to the same sequence motif.

**Binding affinity prediction on randomly generated sequences.** The models generated by machine learning improve binding site prediction (Fig. 1) and distinguish binding events for TFs binding the same motif (Fig. 2). To test if the models are able to produce novel information they were used to predict TF binding to sequences not present in the *A. thaliana* genome. For the HY5 (AT5G11260) TF of the bZIP family with the core motif ACGT, six DNA sequences with high (>150 peak height units) and low (<15 peak height units) regressor binding predictions were generated. For this purpose, 100,000 sequences not present in the genome of *A. thaliana* consisting of 18 bases with ACGT as core sequence were randomly created and the regressor model was applied. Similarly, six DNA sequences were generated for the TF ANAC050 (AT3G10480). The predicted binding affinity was experimentally tested by performing an EMSA (Fig. 3a, b and Supplementary Figs. 16 and 17). Without any competing unlabelled DNA added, a shifted band compared to the negative control indicates TF::DNA binding that is absent upon the addition of unlabelled DNA probe of the same sequence (Fig. 3a, b). In the comparative competition experiment with HY5, adding competing DNA with shapes with low regressor values, all labelled bands are still visible (Fig. 3a). Those shapes are not able to out-compete the labelled sequence and are thus apparently not bound with high affinity by HY5. For the shapes with high regressor values predicted to be bound, two out of three do not show any labelled band and are therefore bound by HY5 with sufficient affinity to out-compete the labelled sequence. For ANAC050 the EMSA shows similar results with five out of six predictions being correct (Fig. 3b). In total, we observed that 10 out of 12 predictions were experimentally validated for both TFs. Given that the AUPRCs for both proteins yielded 0.72 and 0.78 (Fig. 3c, d), the validation of binding and non-binding events occur within the expected error rate. To illustrate the subtle relevant factors, schematic models of the DNA sequences were plotted. For HY5, the schematic model of base and base pair shape shows clear differences on the buckle at position +3 and the shear of position −1 between the bound and not bound sequences (Fig. 3c). Additionally, important positions for binding extracted with SHAP are the helix twist at positions 5 and +1 and the opening at position −1. (Fig. 3c and Supplementary Fig. 18). For the ANAC050 protein, the most obvious difference between the bound and not bound sequence is that the bound sequence is overall more stretched out. The main reason for this observation is that the average roll for the bound sequence is approximately −0.88°, whereas the bases of the sequence which is not sufficiently bound are rolled on average by approximately −1.77° (Fig. 3d). The EMSA confirmed the predictive capability of the models constructed by machine learning.

**Discussion**
Our results show that the binding behaviour of TFs depends on the 3D formation of its binding site, where different TFs favour different formations even within the same protein family. In contrast to ChIP-seq data, ampDAP-seq data, which uses naked

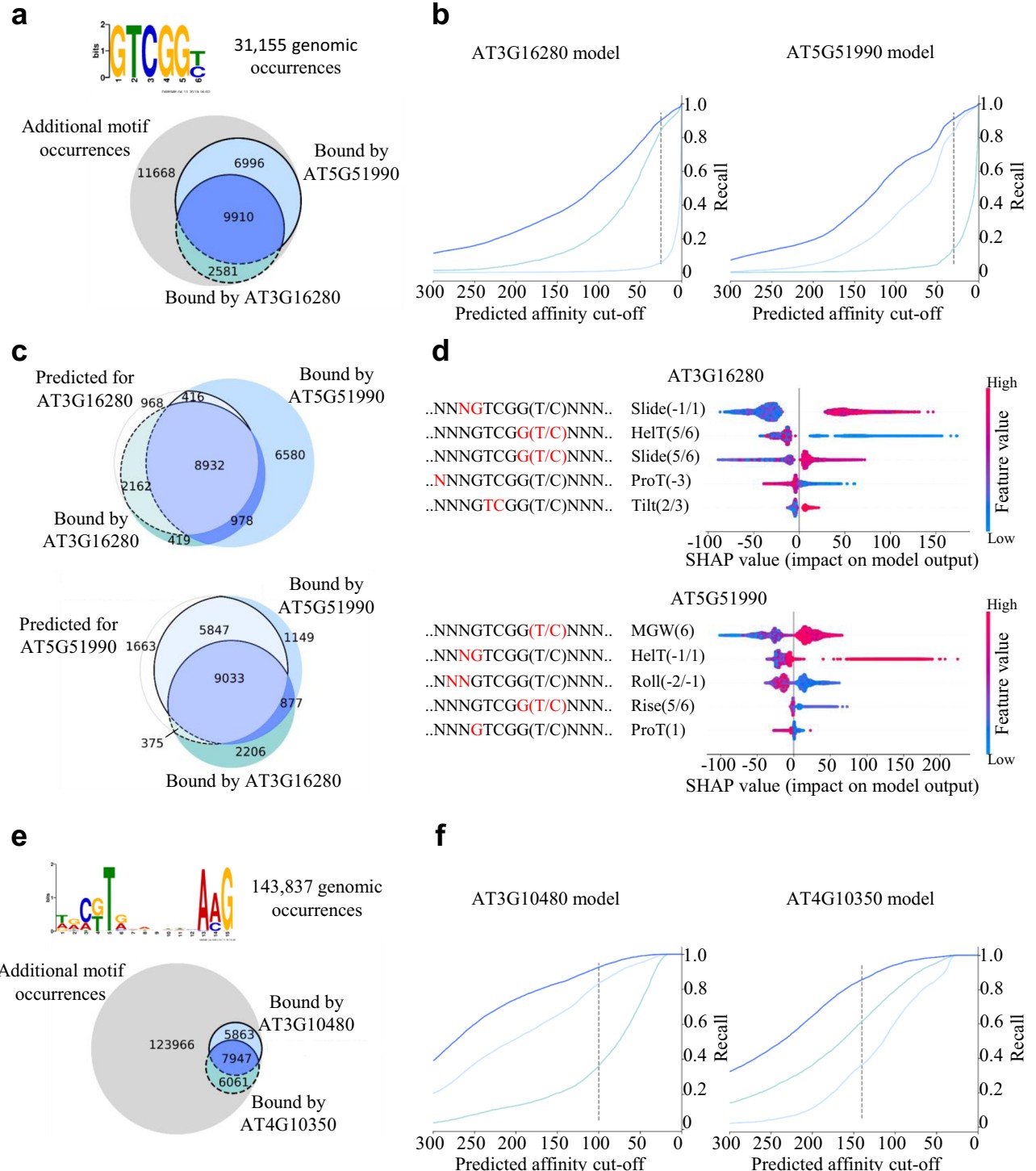

**Fig. 2 Differentiation of binding specificity of intra-familiar proteins with the same binding motif. a, e** Occurrence of the GTCGG(T/C) and C(G/T) TNNNNNNNAAG binding motifs in the *A. thaliana* genome sequence and the experimentally validated binding sequences of the AP2/EREBP TFs AT5G51990 and AT3G16280 and NAC TFs ANAC050 (AT3G10480) and BRN2 (AT4G10350). **b, f** Performance of the random forest regressor trained on the genomic 3D shape. Each line represents the ratio of correctly predicted binding sites regarding all validated binding sites for different affinity prediction cut-offs. The dark blue line corresponds to binding sequences which are bound by both TFs and the light blue lines correspond to the uniquely bound binding sequences. **c** The Venn diagrams show the sequence distributions according to the cut-off represented by the dashed line, respectively. Fields with light colours show the overlap of predicted and validated binding sequences. Dark coloured fields show the quantity of sequences, which were not predicted as bound by the model regarding the shown cut-off. **d** Influence of different local shape features on the prediction of the regressor model. The most influential features are at the top. Each row represents one shape feature at a single position within the sequence.

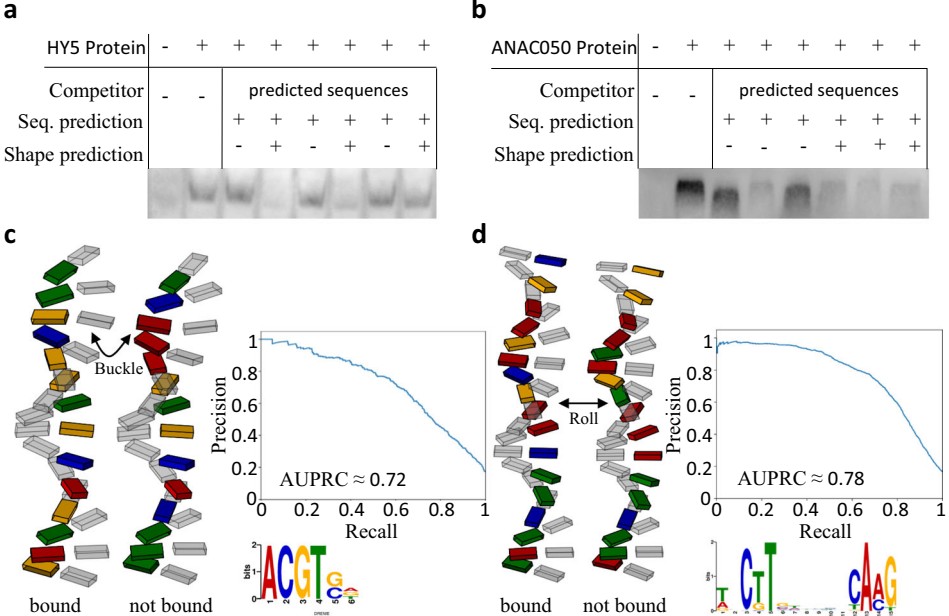

**Fig. 3 Experimental validation of shape-based prediction for HY5 and ANAC050 binding sequences. a, b** Competition EMSA for sequences containing the sequence motif for each respective TF with high and low binding affinity predictions based on their 3D structure. **c, d** Illustration of the 3D structure of the corresponding sequences. The DNA backbone is not shown, as it is not yet possible to reliably calculate the spatial arrangement of the backbone. Additionally, the precision-recall curve of the RF models for the respective TFs are shown. Precision and recall are based on the ampDAP in vitro verified binding sequences.

genomic DNA[4], allows a more precise identification of 3D feature importances for each TF individually. Large experimental efforts have been designed to precisely assign binding sites to TFs[27] and to use this knowledge to describe transcriptional regulation[28]. Our analyses (Figs. 1–3) show that a combination of motif sequence and motif shape enables improved prediction of TF binding on genomic sequence. The models generate a catalogue of potential binding sites in a genome and their predicted affinity. This information forms a base on which additional information layers (i.e. spacing of binding sites[29], chromatin openness[30], histone marks[30], and quantity of TFs and their interactors) can be stacked to enable prediction of gene expression. In synthetic biology, binding events for heterologously expressed TFs can be predicted more precisely, and rationally designed promoter sequences are one step closer.

In the future, it will be critical to study evolutionary trajectories of transcriptional regulation to determine changes to binding sites present in genomes and changes to shape preferences of TFs. Precise understanding of TF binding will allow us to build predictive regulatory networks and hence enable us to understand agriculturally important complex traits, such as differential responses to heat, drought and pathogens and control of yield.

## Methods
**Data processing and extraction.** The ampDAP-seq peak calling data were obtained from the Plant Cistrome Database (neomorph.salk.edu/dap_web/pages/index.php)[4]. Only datasets with a fraction of reads in peaks (FRiP) value >5% were considered for further analyses. All peak sequences were extracted from the *A. thaliana* reference genome sequence (TAIR10), obtained from https://www.arabidopsis.org/. The peak sequences were then used as input for the MEME-ChIP tool[31] to detect binding motifs. The motif with the lowest E-value was chosen as core motif for each TF. Peaks, which appeared in more than one-third of all datasets, are considered as artefacts and were discarded.

To determine motif frequency in the genome, core motif occurrences were searched within the *A. thaliana* genome sequence using FIMO[24]. All motifs located within 80 base pairs of a peak summit were considered as experimental validated binding events. Multiple motif occurrences within this defined peak area were classified as homodimer binding sites to enable a more precise signal value

interpretation. The calculation of the DNA shape was performed using a publicly available query table[6,32] provided from https://rohslab.usc.edu/DNAshape+/.

The RF classifier as well as the RF regressor models were generated and trained using the python module scikit-learn[25]. Hyperparameter grid search and 5-fold cross-validation were performed to generate each model. A more detailed explanation of the data pre-processing and model generation is provided in the subsection below. Code is available from GitHub (https://github.com/janiksielemann/shape-based-TF-binding-prediction). Required python packages are pandas[33], numpy[34], scikit-learn[25], biopython[35], matplotlib[36], shap[26], scipy[37] and dabest[38].

**Pre-processing and training of the random forest regressor.** To perform data pre-processing and training of a random forest model an ampDAP-seq peak file (from http://neomorph.salk.edu/dap_web/pages/browse_table_aj.php) and the *A. thaliana* genome (from arabidopsis.org) is necessary. Only peak files with FriP value >5% were considered and after motif extraction each peak file was filtered for peaks that appear in less than 66% of all ampDAP-seq available peak files, as those peaks were considered artefacts due to the ampDAP-seq procedure. Peak regions were extracted from the Arabidopsis genome using a custom python script, which expects one peak file (-p) and the corresponding genome (-g) as input. The resulting fasta file with genomic peak regions was used as input for the MEME-ChIP[31] (MEME-suite[20] v 5.0) tool with default parameters, so that the only given parameters were an output folder (-oc) and the peak regions fasta file (-dna). The sequence motif with highest E-value (--motif 1) from the resulting combined.meme file was searched in the *A. thaliana* genome using the FIMO[24] (MEME-suite[20] v 5.0) tool with a cut-off (--thresh) of 5e-4. To ensure that no matches were discarded the maximum number of stored matches --max-stored-scores) were set to 1,000,000. The parameter --max-strand was set to 1 so that palindromic sequences would not match two times in the same locations and an output folder (--oc) was declared.

To allow a more accurate interpretation of binding affinities, the areas of motif matches were scanned for multiple motif occurrences using a custom python script. For this, each peak, which always has a length 200 base pairs, was tested for multiple FIMO matches. If more peaks had multiple motif occurrences than single motif occurrences, the corresponding TF was considered for homodimeric binding events and vice versa. In that case, the random forest regressor was only trained on those peaks with multiple motif occurrences.

All genomic locations with sequence motif matches were translated into 13 DNA shape features using a publicly available query table[6], which was implemented into a custom Python script. Chloroplast and mitochondrial motif occurrences were discarded, as those sequences were not part of the in vitro experiment. Additionally, all sequences that were initially hit on the reverse strand were reverse complemented. The sequence window, for which the DNA shape was calculated, was set to 32 additional bases upstream and downstream from the

sequence motif. Experimentally recorded signal values were normalised to range from 0 to 1000 using sklearn pre-processing module[25]. Since some shape features are mirrored for palindromic sequences, each matched sequence window on the minus strand was reverse complemented, so that the matrix of 3D shape values always correspond to the same direction.

DNA shape-based training to learn protein binding affinities was performed with the RandomForestRegressor class from the sklearn module. The number of considered positions upstream and downstream of the sequence motif can be specified using the custom Python script by setting the -b parameter, which has a default value of 4. If the ratio of experimentally verified binding sites and genomic binding site occurrences was too high (>1:5), this ratio was forced to be 1–5 by discarding random genomic positions. If this procedure would still yield more than 120,000 locations, the ratio was forced to be 1–3. This ratio between validated binding sites and binding site occurrences was always calculated for each TF and used as sample weight for training the random forest regressor to prevent bias towards false negatives. The whole dataset was split into a train set (80%) and a validation set (20%) using the train_test_split function provided by the sklearn module, applying stratification to ensure even distributions of validated binding sites in the train and test set. The train set was used for 5-fold cross-validation learning and the validation set was used for evaluation. Within the 5-fold cross-validation, the train set was split into train and test set with a ratio of 4:1. The model was then trained five times so that each instance was part of the test set once. To finetune the learning process, randomised hyperparameter grid search was performed for 75 iterations including the parameters "n-estimators" (ranging from 10 to 200), "max_features" (auto, square root or log2) and "max_depth" (ranging from 4 to 12). As each of the 75 iterations for hyperparameter tuning was 5-fold cross-validated as described, a total number of 375 training procedures for each respective TF was performed. The mean squared error was used as loss. For evaluation purposes the precision recall curve function, which is also provided by the sklearn module, was applied on the validation data. The validation data was not used for hyperparameter tuning but solely for evaluation, as it was separated from the train data within pre-processing. After training the model, sequences of interest can be checked for putative binding affinity.

**Training of other baseline models**. To perform the comparative analysis between machine learning approaches we trained gradient boosting models and neural networks for each TF, respectively. The pre-processing steps were the same for each approach, so that each approach had the same input dataset.

For the gradient boosting approach we used the "GradientBoostingRegressor" class from the sklearn python package. Besides setting a random state, the default parameters were used for the baseline model.

To build the neural network model, the "Sequential" class from the keras API was used. The input shape was defined according to the number of input features for the respective TF, as the length of the core motif differed from protein to protein. A dense layer with 200 neurons and ReLU as activation function was added, as well as an output layer to predict the signal value. The model was compiled, defining the mean squared error as loss and keras "Adam" class as optimiser with a learning rate of 0.001.

**Example application based on the transcription factor HY5**. For the TF HY5, the peak calling of the in vitro ampDAP-seq experiment identified 10,140 DNA-binding sites (Fig. 1a). Those binding sites were used to calculate the sequence motif using MEME-ChIP[31]. We referred to the resulting sequence motif as "core motif", as we took additional bases upstream and downstream from the motif to convert the sequence into shape features as described. Using this sequence motif of HY5 to extract all genomic sequences that contain the motif yields 55,740 genomic sequences (Fig. 1a). The mitochondrion, as well as the chloroplast were not considered.

The extracted 55,740 potential HY5 binding sequences were converted to DNA shape features[6]. In the case of HY5, four bases upstream and downstream of the core motif were incorporated to convert the sequence into shape features. As shown in the illustration, additional pre-processing steps like the calculation of the sample weights were conducted. All samples were labelled according to the measured signal value within the peak calling of the ampDAP-seq experiment to enable the regression task.

A validation set which contained 20% of the dataset was separated to ensure an independent evaluation. This dataset was not used for hyperparameter tuning. The random forest regression model was trained on the remaining 80% of the dataset, which was again split into train and test set within the 5-fold cross-validation procedure (Fig. 1a). For hyperparameter tuning a grid search was performed. The best performing model for HY5 ended up with the parameters of 'n_estimators' = 190, 'max_features' = 'sqrt' and 'max_depth' = 11. This model was used to predict the signal values of the samples in the validation set and the performance was evaluated by calculating a precision recall curve.

**Experimental procedure**. The HY5 (AT5G11260) and ANAC50 (AT3G10480) coding sequences were cloned with Gibson assembly in pFN19A HaloTag® T7 SP6 Flexi® Vector (Promega, Madison, WI, USA; Cat.: G921A; Batch: 0000341144; 1:10,000) in an N-terminal fusion with the Halo-tag. Plasmid DNA was isolated with the

ZymoPURE Plasmid Midiprep kit (ZymoGenetics, Seattle, WA, USA). The HY5 protein was expressed with TnT® SP6 High-Yield Wheat Germ Protein Expression System (Promega, Madison, WI, USA) using 2 μg plasmid DNA per 50 μL expression reaction. The ANAC050 protein was purified with the HaloTag® Protein Purification System (Promega, Madison, WI, USA) using 20 μL expression reaction for each EMSA reaction. Expression was validated by Halo-tag detection (Supplementary Fig. 14). Double-stranded DNA sequences (20 μM) were generated by annealing synthesised DNA (98–21 °C, 9 h) and diluted to 0.25 μM. The binding reaction was incubated for 2 h at 21 °C. A 5% native polyacrylamide gel containing 0.5 TBE and 2.5% glycerol was pre-run for 30 min. The samples were loaded with 1 μL orange loading dye (Thermo Fisher Scientific, Waltham, MA, USA) and the gel (10 × 7.5 cm) was run at 80 V until the OrangeG front was 1 cm before the end of the gel. The gel was blotted on a positively charged nylon membrane (Hybond™, GE Healthcare, Chicago, IL, USA) at fixed current of 0.8 mA/cm² for 90 min. The DNA was fixed by UV for 10 min. Biotin labelled DNA was detected with 1:5000 solution of an anti-biotin HRP-conjugated antibody (BioLegend, San Diego, CA, USA; Cat.: 405210; Batch: B293545; 1:5000) in TBST with 5% BSA. Detection was performed using Pierce™ ECL Western Blotting Substrate (Thermo Fisher Scientific, Waltham, MA, USA) as described by the manufacturer and the imaging system Fusion Fx7 (Vilber, Collégien, France).

**Reporting summary**. Further information on research design is available in the Nature Research Reporting Summary linked to this article.

## Data availability
Source data are provided with this paper, containing the sequences used for the EMSA, uncropped gel images and resulting values, which were used to create the figures. To translate DNA sequence into shape features, the publicly available query table (https://rohslab.usc.edu/DNAshape+/) was used[6]. The ampDAP-seq peak calling data, which were used as ground truth to train the models, were obtained from the Plant Cistrome Database (neomorph.salk.edu/dap_web/pages/index.php). Source data are provided with this paper.

## Code availability
The code to train a model and predict binding affinities for a given transcription factor is available from GitHub (https://github.com/janiksielemann/shape-based-TF-binding-prediction)[39].

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

## Acknowledgements
We thank the Bioinformatic Resource Facility team at the Center for Biotechnology (Bielefeld University) for technical support. J.S. is funded by the Digital Infrastructure in the Life Sciences graduate school (Bielefeld University). D.W. is supported by core funding, Bielefeld University.

## Author contributions
J.S. designed and carried out the computational experiments including programming, interpreted the data and co-wrote the paper. D.W. designed and carried out the wet lab experiments, interpreted data and edited the paper. R.S. assisted with the wet lab experiments, interpreted data and edited the paper. A.B. conceived the initial idea and the study, interpreted data and co-wrote the paper.

## Funding

## Competing interests
The authors declare no competing interests.
