## [Peer Review File. · Nature Communications]

Local DNA shape is a general principle of transcription factor binding specificity in *Arabidopsis thaliana*REVIEWER COMMENTS

Reviewer #1 (Remarks to the Author):

In this manuscript, the authors tested the idea that binding of a transcription factor (TF) to a cis-regulatory region is determined not only by the core sequence motif, but also by the three-dimension (3D) structure of the bases surrounding the core motif. Moreover, while the sequence of core motif determines binding of the same TF family, the 3D information of the surrounding bases determines the binding specificity of the different members in the same family. This conclusion was computationally validated by prediction analysis of TF binding affinity: when both motif sequence and 3D information were considered by the predictive model, the accuracy was significantly improved compared to only using sequence motif for prediction. The rationale of this work is absolutely novel and interesting, but technical aspects might be problematic, considering the small sample size (only 216 TFs) and the machine learning (ML) method the authors used for prediction. I raised the below concerns and advices for authors' further improvement.

Major issues:

1. Organization of the manuscript is problematic. The Abstract part is lengthy, but mostly presents common knowledge regarding transcription regulation and the novelty of this work was not well delivered. The same problem occurs for the Introduction part. Many results are worthy of an in-depth discussion, but the authors neglected. Writings and Figures are of low quality, as well as many typos and mistakes through the manuscript. Descriptions of the methods, especially for the procedure of using ML, were not detailed enough, so that I cannot really judge whether the method was properly used.

2. The selling point was the utilization of Random Forests (RF) algorithm to build the model for prediction of TF binding sites. However, this is probably the most problematic part, making the results inconvincible. ML is a powerful tool for data mining and prediction, but if it was not appropriately used, it will suffer serious overfitting issue. The characteristics of this dataset is the sample size of the 216 TFs is much smaller than the feature set. Therefore, a simple 5-fold cross validation is not sufficient to rule out the possibility of overfitting, and the authors need to performs many ways of testing as follows. 1) The dataset should be partitioned to training, testing and validating set. While training and testing were used for tuning model, the validating set has to be absolutely independent and the procedure needs to be repeated for at least ten times to generate a distribution to evaluate accuracy. Additionally, the ratio of training and testing (4:1) may be changed to 3:1, 2:1 and 1:1 to evaluate the influence of sample size on accuracy. 2) The authors compared the results of motif search-only and motif search+3D shape. What the results would look like, if only 3D shape was used as features for prediction. This would be the baseline accuracy. 3) The authors may shuffle the features of 3D shapes or generated a simulated feature set with equal amount of the actual features. This is similar to a permutation test, so that we

may quantitatively judge the actual effectiveness of the 3D shape as features. 4) If the features are excessively redundant, dimensionality-reduction on features is necessary. 5) If possible, comparison of multiple ML models, such as SVM, artificial neural network, gradient boosting tree, i.e., may be compared in parallel to determine the best model.

3. Figure S4 showed that the improvement of prediction accuracy varied in different families of TFs when adding 3D shape as extra features. Is it related to the different number of samples in different family? The same question for Figure S5, in which the different prediction performance of the binding specificity is related to the conservation of the binding domain in the TF proteins?

4. Table S1 listed the top 5 features of 3D shape for each TF. This result may be used to test the effectiveness of feature selection, and may be also used to rule out the possibility of overfitting. For example, select 2/3 of TFs in each family as training set and use the RF for feature importance evaluation when training model. Then, the top five or top ten features were used for RF to predict TF binding site in the 1/3 TFs in the testing set. If the prediction remains high, it means the model is robust and stable.

Reviewer #2 (Remarks to the Author):

The authors' have developed a random forest classifier based on DNA shape properties to improve the accuracy of prediction of DAP-seq binding sites and intensity compared to the use of motif alone. While I think the concept is interesting and likely to be valuable, and computational approaches appear sound, my major concern is that they have chosen to use "core motif" instead of the actual motif/PWM from the DAP-seq data. For example, in Figure 2 they use "GTCGGT/C" as a core motif for AT5G51990 and AT3G16820. While both of these factors do share this core motif, both also have flanking sequence on either side of the core motif that are distinct (not shown in their figure). Though these "edge" motif bases contribute less to the overall bit score than the core motif, they are almost certainly going to contribute to differences in the selectivity of the TF binding for these two factors. Thus, in Figure 2A Venn diagram there really should be two grey circles, one for targets of AT5G51990 and one for AT3G16820. I would not expect them to completely overlap. In fact, the shape predictor may be capturing some of these same edge motif bases. As the authors show in Figure 1D the majority of the AUPRC gains are in the core +2-4bp, where these additional edge motif sites would be located. Similarly, in Figure 2E the two NAC factors both have very distinct bases on their edges, and by not including those it's difficult to know how much of the algorithm is recovering shape features or simply capturing these extra edge bases that were left out of the analysis.

On the positive side, some of the analysis in Figure 2 and Figure 3 showing the potential role at individual bases in the core as well as flanking regions to influence the shape of the DNA can shed light on how sequence-to-shape influences binding affinity. Additionally, if full motifs are used and the authors can capture additional information with their model to further improve prediction this could be a useful tool for improved de novo prediction in other plant species which do not have DAP-seq sequence. However, the use of only “core” motifs makes it impossible at this point to determine what these benefits may be.

Smaller points:

1. Why did they chose to use ampDAP instead of the DAP-seq data. This isn't explained.
2. Some language may be improved. For example, they use the phrase “not-in-genome” line 25 of the abstract which was confusing. Later they define it as a set of simulated sequences. Perhaps that would be a better way to describe it. Another example is they use the phrase “histone flags” which I am more commonly familiar with the phrase “histone marks”.

Reviewer #1 (Remarks to the Author):

In this manuscript, the authors tested the idea that binding of a transcription factor (TF) to a cis-regulatory region is determined not only by the core sequence motif, but also by the three-dimension (3D) structure of the bases surrounding the core motif. Moreover, while the sequence of core motif determines binding of the same TF family, the 3D information of the surrounding bases determines the binding specificity of the different members in the same family. This conclusion was computationally validated by prediction analysis of TF binding affinity: when both motif sequence and 3D information were considered by the predictive model, the accuracy was significantly improved compared to only using sequence motif for prediction. The rationale of this work is absolutely novel and interesting, but technical aspects might be problematic, considering the small sample size (only 216 TFs) and the machine learning (ML) method the authors used for prediction. I raised the below concerns and advices for authors' further improvement.

Major issues:

1. Organization of the manuscript is problematic. The Abstract part is lengthy, but mostly presents common knowledge regarding transcription regulation and the novelty of this work was not well delivered. The same problem occurs for the Introduction part. Many results are worthy of an in-depth discussion, but the authors neglected. Writings and Figures are of low quality, as well as many typos and mistakes through the manuscript. Descriptions of the methods, especially for the procedure of using ML, were not detailed enough, so that I cannot really judge whether the method was properly used.

Thank you for this constructive concern. We agree that we did not focus enough on data pre-processing and the method itself beyond the github in which we deposited all necessary scripts. In this new version, we added a visualisation, illustrating the workflow based on one example transcription factor (Figure 1A, Supplemental Methods Figure 1) to the manuscript. We also added in-depth explanations about the machine learning procedure in the supplementary methods and explain what happens at each step of the pipeline.

The figure quality might appear low in the reviewed version of the manuscript. We made sure that all figures have a resolution of 300 DPI. The manuscript was again thoroughly revised to eliminate typos, improve language and restructure the manuscript.

We cut large parts of the abstract and rewrote most of it to enable a more precise description of the topic and our findings and create space to more precisely explain data processing and the method in the manuscript.

We added more detailed discussion parts to some of the findings to elaborate on the expectations and implications of those observations. For example, we added lines 140-143:

“This observation is in line with our expectation, as ampDAP-seq identifies binding events independent from conditions in the cell and uses unmethylated naked DNA for the identification of binding sites, whereas ChIP-seq captures the binding events in specific in vivo conditions.”

2. The selling point was the utilization of Random Forests (RF) algorithm to build the model for prediction of TF binding sites. However, this is probably the most problematic part, making the results inconvincible. ML is a powerful tool for data mining and prediction, but if it was not appropriately used, it will suffer serious overfitting issue. The characteristics of this dataset is the sample size of the 216 TFs is much smaller than the feature set. Therefore, a simple 5-fold cross validation is not sufficient to rule out the possibility of overfitting, and the authors need to performs many ways of testing as follows. 1) The dataset should be partitioned to training, testing and validating set. While training and testing were used for tuning model, the validating set has to be absolutely independent and the procedure needs to be repeated for at least ten times to generate a distribution to evaluate accuracy. Additionally, the ratio of training and testing (4:1) may be changed to 3:1, 2:1 and 1:1 to evaluate the influence of sample size on accuracy. 2) The authors compared the results of motif search-only and motif search+3D shape. What the results would look like, if only 3D shape was used as features for prediction. This would be the baseline accuracy. 3) The authors may shuffle the features of 3D shapes or generated a simulated feature set with equal amount of the actual features. This is similar to a permutation test, so that we may quantitatively judge the actual effectiveness of the 3D shape as features. 4) If the features are excessively redundant, dimensionality-reduction on features is necessary. 5) If possible, comparison of multiple ML models, such as SVM, artificial neural network, gradient boosting tree, i.e., may be compared in parallel to determine the best model.

We apologize for the apparently insufficient description of the method in the initial manuscript. Thank you for this detailed concern and the proposed improvements. The main concern is that the model is vulnerable to overfitting due to low sample size and a large feature set. We would like to explain that not a single model is trained on the 216

transcription factors but rather one model is trained for each transcription factor, respectively. Additionally, the model is trained using all genomic sequences that contain the sequence motif of the respective transcription factor. Thus, the sample size in fact is much larger than the feature set. We clarify the input data and the ML approach in general with a more detailed explanation and visualisation in the supplementary methods as well as Figure 1A .

- 1) For each of the 216 trained models, 20% of the input data was separated and only used for evaluation. In the manuscript this data is called validation set and it was not used for hyperparameter tuning. The remaining 80% of the input data, which we called train set, was used for hyperparameter tuning using 5-fold cross validation and grid search. Within the cross-validation process, the train set is split again into train and test set with ratio of 4:1. We elaborate on this in by adding explanations and a visualisation to the supplementary methods. We addressed the suggestion to change the train and test ratio by training each model with train-test ratios of 3:1, 2:1 and 1:1 respectively, so that 648 (3*216) new models were trained and evaluated. We added a supplementary figure (Supplemental Figure 3) to the manuscript and interpreted the results. We added this paragraph in lines 115-119:

“The training dataset was again split into train and test set (ratio 4:1) while performing cross validation. In addition, we explored different ratios of train to test set (4:1, 3:1, 2:1 and 1:1) and observed no difference in performance (Supplemental Figure 3). This indicates that the size of the input dataset is sufficient for robust training.”

- 2) The sequence motif was used to extract all genomic sequences that contain the sequence motif of the respective transcription factor, but it was not used as input feature for the model. The “motif search-only” approach uses the similarity between the sequence motif and the extracted genomic sequence which is calculated by FIMO. Our models use only the shape features of the sequence as input in the “motif search+3D shape” approach. This context was poorly described in the original manuscript. Both approaches use the sequence motif to extract the genomic sequences which contain the motif. We clarify this process by adding figure 1A and elaborating on the workflow in the supplementary methods.
- 3) As discussed in 2), our models solely use the 3D shape features as input for training and prediction to separate binding from non-binding events.
- 4) To address this concern, we performed PCA on the input sequence shapes for each transcription factor, respectively. There seems to be little redundancy using the 3D shape features, as we observed low percentages of explained variance in the first principal components. We trained all models with the dimensionally reduced feature

sets for each transcription factor. This is shown in the new supplemental figure (Supplemental Figure 4) and we discussed the observation in lines 126-130:

“To investigate the influence of dimensionally reduced input features on model performance, all models were additionally trained after performing PCA on the shape features (Supplemental Figure 4). The prediction performance was substantially lower when the models were trained on the dimensionally reduced features rather than the direct shape features. This observation also implies that the features are not considerably redundant.”

- 5) For each transcription factor we now additionally trained baseline models using gradient boosting and a neural network. The implementation is now described in the supplemental methods. We evaluated the approaches by comparing the area under the precision recall curve of the predictions for the validation set, which was, as described, not seen by the models before. We added a supplemental figure (Supplemental Figure 10) to the manuscript and discussed the resulting evaluation in lines 149-152:

“Two additional machine learning approaches were evaluated and compared with RF performance (Supplemental Figure 10). The baseline neural network implementation performed overall slightly worse than the gradient boosting and random forest implementation. This relation would likely change with dedicated hyperparameter tuning.”

3. Figure S4 showed that the improvement of prediction accuracy varied in different families of TFs when adding 3D shape as extra features. Is it related to the different number of samples in different family? The same question for Figure S5, in which the different prediction performance of the binding specificity is related to the conservation of the binding domain in the TF proteins?

The number of samples for each transcription factor is dependent on its sequence motif, as the samples are all genomic sequences which contain the respective motif. A unique model is trained for each transcription factor, so that in the end we have 216 models which were trained on differently sized datasets, depending on the number of extracted genomic sequences. It is indeed interesting to investigate whether the performance of the model correlates with the dataset size. Thus, we calculated the correlation between dataset size and performance and added a supplemental figure (Supplemental Figure 7). We discussed the findings lines 132-137:

“We analysed whether this discrepancy could be explained by the different dataset sizes, which is the number of genomic sequences containing the motif (Supplementary Figure 7). Indeed, we observed a slightly negative correlation between performance and dataset size with an R^2 of 0.154. Hence, on average the prediction performance is slightly worse for TFs whose sequence motif is more abundant in the genome.”

4. Table S1 listed the top 5 features of 3D shape for each TF. This result may be used to test the effectiveness of feature selection, and may be also used to rule out the possibility of overfitting. For example, select 2/3 of TFs in each family as training set and use the RF for feature importance evaluation when training model. Then, the top five or top ten features were used for RF to predict TF binding site in the 1/3 TFs in the testing set. If the prediction remains high, it means the model is robust and stable.

Thank you for this interesting suggestion. As our approach trains one model for each transcription factor respectively, this is not implementable using our approach. Yet though, if a single model would be trained for binding affinity prediction considering all transcription factors, this would be a good way to ensure the robustness of the model.

Reviewer #2 (Remarks to the Author):

The authors' have developed a random forest classifier based on DNA shape properties to improve the accuracy of prediction of DAP-seq binding sites and intensity compared to the use of motif alone. While I think the concept is interesting and likely to be valuable, and computational approaches appear sound, my major concern is that they have chosen to use “core motif” instead of the actual motif/PWM from the DAP-seq data. For example, in Figure 2 they use “GTCGGT/C” as a core motif for AT5G51990 and AT3G16820. While both of these factors do share this core motif, both also have flanking sequence on either side of the core motif that are distinct (not shown in their figure). Though these “edge” motif bases contribute less to the overall bit score than the core motif, they are almost certainly going to contribute to differences in the selectivity of the TF binding for these two factors. Thus, in Figure 2A Venn diagram there really should be two grey circles, one for targets of

AT5G51990 and one for AT3G16820. I would not expect them to completely overlap. In fact, the shape predictor may be capturing some of these same edge motif bases. As the authors show in Figure 1D the majority of the AUPRC gains are in the core +2-4bp, where these additional edge motif sites would be located. Similarly, in Figure 2E the two NAC factors both have very distinct bases on their edges, and by not including those it's difficult to know how much of the algorithm is recovering shape features or simply capturing these extra edge bases that were left out of the analysis. On the positive side, some of the analysis in Figure 2 and Figure 3 showing the potential role at individual bases in the core as well as flanking regions to influence the shape of the DNA can shed light on how sequence-to-shape influences binding affinity. Additionally, if full motifs are used and the authors can capture additional information with their model to further improve prediction this could be a useful tool for improved de novo prediction in other plant species which do not have DAP-seq sequence. However, the use of only "core" motifs makes it impossible at this point to determine what these benefits may be.

Thank you for the detailed concern and suggestions. To generate the core motif we used all binding events from the DAP-seq peak calling. The published motif within the DAP-seq data was calculated by using the top 600 binding events. Those sequence motifs most certainly represent the sequence with the strongest affinity, although most of the binding events are neglected for the generation of the sequence motifs. We clarified our procedure to generate the core motifs in the supplemental methods and added lines 90-94 in the supplemental methods:

"For the transcription factor HY5, the peak calling of the in vitro ampDAP-Seq experiment identified 10,140 DNA binding sites (Supplemental methods figure 1). Those binding sites were used to calculate the sequence motif using MEME-ChIP¹. We referred to the resulting sequence motif as "core motif", as we took additional bases upstream and downstream from the motif to convert the sequence into shape features as described."

Based on the suggestions, we compared the extracted genomic locations when using either motif. We observed that, contrary to expectations, the usage of the published motif from the top 600 peaks leads on average to more extracted genomic sequences compared to the motif derived from all peaks. This is most likely due to longer motifs surpassing the p-value threshold for identification easier than shorter motifs. We added a supplemental figure (Supplemental Figure 2) and discussed this observation in lines 98-106:

"To generate the datasets necessary for training, test, and validation, for each TF the sequence-based binding motif (henceforth called "core motif") was determined with MEME-chip using all ampDAP-seq peaks. Sometimes a motif is reported based on only the 600

peaks with the largest height; we opted to capture all binding events. The genome was scanned with the motif generating two classes of events: motifs which are not underneath a peak and hence not bound and motifs underneath a peak which were bound. Peak height is taken as a proxy for affinity. If a motif based on only the top 600 peaks was used, the number of sequence-only based potential binding sites was increased (Supplemental Figure 2) likely because those larger motifs reach the threshold for FIMO24-based extraction more easily compared to smaller motifs.”

Additionally, the proposed analysis was carried out, for which we compared the overlaps for AT5G51990 and AT3G16280 of extracted genomic sequences using the published “top 600 peaks motif”. As discussed, this approach finds more genomic occurrences, namely 103,779 locations in contrast to the previous 31,155 locations for both transcription factors. We still observe a major overlap, but, as you suspected, especially for AT5G51990 a noticeable part of the binding locations is found only in the corresponding extracted motif locations. We added a supplemental figure (Supplemental Figure 12) and added lines 175-177:

“Using the published motif derived from the top 600 binding events results in a smaller overlap but leads to 103,779 extracted genomic sequences (Supplementary Figure 12).”

Minor issues:

1. Why did they chose to use ampDAP instead of the DAP-seq data. This isn't explained.

We chose ampDAP data over DAP data because we wanted to use binding events on unmethylated naked DNA as ground truth for our input dataset. It is known that methylation influences protein-DNA binding and this should clearly be considered when trying to deduce DNA binding for biotechnological applications. Yet though, for this study we decided to use data which is as independent as possible from molecular modifications of DNA. See lines 60-61:

“In amplified DNA affinity purification sequencing (ampDAP-seq) experiments amplified DNA devoid of methylation marks is bound to an in vitro produced TF and sequenced.”

2. Some language may be improved. For example, they use the phrase “not-in-genome” line 25 of the abstract which was confusing. Later they define it as a set of simulated sequences.

Perhaps that would be a better way to describe it. Another example is they use the phrase “histone flags” which I am more commonly familiar with the phrase “histone marks”.

Thank you for the suggestions. We revised the manuscript and endeavoured to improve the language throughout. The addressed phrase was changed:

From “*not-in-genome motif sequences*” to “*randomly generated motif containing sequences*” and the phrase “*histone flags*” was changed to “*histone marks*” in lines 36 and 272.

REVIEWERS' COMMENTS

Reviewer #1 (Remarks to the Author):

The authors have fully addressed my previous concerns, especially my second question which involves lots of work. The newly added analyses greatly improved the manuscript, and I have no further scientific questions.

My last request is that all of the codes, data related to the main analysis and results in the figure has to be uploaded to zenodo with a assigned DOI included in the paper. Then readers interested in this work may repeat the analysis. It's not sufficient to only put main program on Github. This is better done before the official acceptance.

Reviewer #2 (Remarks to the Author):

The authors' changes and clarifications have satisfied all of my concerns.

REVIEWERS' COMMENTS

Reviewer #1 (Remarks to the Author):

The authors have fully addressed my previous concerns, especially my second question which involves lots of work. The newly added analyses greatly improved the manuscript, and I have no further scientific questions.

My last request is that all of the codes, data related to the main analysis and results in the figure has to be uploaded to zenodo with a assigned DOI included in the paper. Then readers interested in this work may repeat the analysis. It's not sufficient to only put main program on Github. This is better done before the official acceptance.

Thank you for this advice. We updated the GitHub repository to now also include a download script for raw data extraction to enable better reproducibility. Additionally, we added used scripts beyond the main analysis. We also uploaded the repository to Zenodo to assign a DOI to the current version of the scripts and added a citation in the "Code availability" section.

Reviewer #2 (Remarks to the Author):

The authors' changes and clarifications have satisfied all of my concerns.